# Screening for Antimicrobial Resistance and Genes of Exotoxins in *Pseudomonas aeruginosa* Isolates from Infected Dogs and Cats in Poland

**DOI:** 10.3390/antibiotics12071226

**Published:** 2023-07-24

**Authors:** Daria Płókarz, Karolina Bierowiec, Krzysztof Rypuła

**Affiliations:** Division of Infectious Diseases of Animals and Veterinary Administration, Department of Epizootiology and Clinic of Birds and Exotic Animals, Faculty of Veterinary Medicine, Wroclaw University of Environmental and Life Sciences, pl. Grunwaldzki 45, 50-366 Wroclaw, Poland; daria.plokarz@upwr.edu.pl (D.P.); karolina.bierowiec@upwr.edu.pl (K.B.)

**Keywords:** *Pseudomonas aeruginosa*, pets, antibiotic resistance, virulence-factor-encoding genes, exotoxins, disc diffusion method

## Abstract

*Pseudomonas aeruginosa* has assumed an increasingly prominent role as the aetiological agent in serious hard-to-treat infections in animals and humans. In this study, 271 *P. aeruginosa* strains collected from dogs and cats were investigated. The aim of the research was to screen these *P. aeruginosa* strains for antibiotic resistance and the presence of selected virulence factor genes. Antibiotic resistance was determined using the Kirby–Bauer method, while virulence genes were detected by polymerase chain reaction (PCR). The most frequently detected resistance was to fluoroquinolones, ranging in prevalence from 17.3% for ciprofloxacin up to 83% for enrofloxacin. The resistance to carbapenems was 14% and 4.8% for imipenem and meropenem, respectively. Almost all *P. aeruginosa* strains harboured the *exoT* (97.8%) and *lasB* (93.4%) genes, while the lowest prevalence was found for *exoU* (17.3%) and *plcH* (17.3%). *P. aeruginosa* strains isolated from dogs that harboured the *toxA* gene were more frequently resistant to ceftazidime (*p* = 0.012), while the presence of the *exoU* gene was found to be connected with resistance to marbofloxacin (*p* = 0.025) and amikacin (*p* = 0.056). In strains originating from cats, only the connection between the presence of the *exoU* gene and resistance to enrofloxacin (*p* = 0.054) was observed. The confirmation of associations between virulence-factor-encoding genes and antibiotic resistance indicates that problems of antibiotic resistance may not only cause complications at the level of antibiotic dosage but also lead to changes in the virulence of the bacteria; thus, further studies in this area are required.

## 1. Introduction

*Pseudomonas aeruginosa* (*P. aeruginosa*) is an oxidase-positive Gram-negative bacillus with minimal nutrient requirements, a feature that, combined with the possibility of its growth across a broad range of temperature (4–44 °C), is recognized to contribute to its ubiquity [1]. In nature, *P. aeruginosa* can exist in biofilm, attached to a surface or substrate, or in planktonic form, in which it can actively move via its flagellum [2]. *P. aeruginosa* is an opportunistic bacterium and very rarely causes infection in healthy individuals; however, in certain circumstances, it can directly threaten the health or the life of an individual [1,3,4]. Nevertheless, due to its high resistance level to antimicrobials and disinfectants, *P. aeruginosa* is a major cause of nosocomial infections in humans (7.1% in European countries) [5,6]. The risk factors to humans associated with *P. aeruginosa* infections are chronic obstructive pulmonary disease, diabetes, cystic fibrosis, immunosuppression (after organ or bone marrow transplantation), severe kidney and liver failure, and multiorgan failure [7,8,9]. In dogs the most frequent *P. aeruginosa* is isolated from patients with ear infections, nevertheless, the bacteria can cause a range of other infections including skin infections, urinary tract infections, and respiratory infections [10,11,12]. While in cats *P. aeruginosa* is mostly described in clinical material from wounds, respiratory tract infections, and conjunctivitis [13,14].

The diversity of the antimicrobial mechanism of *P. aeruginosa* causes difficulties in eradicating the bacteria from healthcare environments. The outer membrane of *P. aeruginosa* contains a variety of porins that affect movement in and out of the cell and efflux pumps that can actively pump antimicrobials out. Moreover, compared with other Gram-negative bacteria, it is less permeable to antibiotics [15]. *Pseudomonas* are naturally resistant to penicillin G, aminopenicillin, cephalosporin of the I and II generation, macrolides, tetracyclines, chloramphenicol, quinolones, sulphonamides, and trimethoprim [1], which narrows down the available clinical options, especially for veterinary patients [16]. In the case of *P. aeruginosa*, there is always a risk to public health [17]. According to the European Food Safety Authority, Panel on Animal Health and Welfare, *P. aeruginosa* is classified, with >90% certainty, as one of the most relevant antimicrobial-resistant bacteria in the EU [18].

The possibility of antimicrobial-resistant strain transfer from dogs and cats to humans should be an important point in international infectious disease control [19]. Particular attention within the context of public health is given to the carbapenem resistance of *P. aeruginosa* due to its increased prevalence outside of hospital environments. Carbapenems, such as imipenem and meropenem, are effective antibiotics for the treatment of *P. aeruginosa* infections [20]. In recent years, the rate of carbapenem resistance in *P. aeruginosa* has increased worldwide and has become of great concern, since it significantly restricts the therapeutic options for patients. Increased carbapenem resistance among these organisms has been found worldwide, with around 10% to 50% of *P. aeruginosa* isolates recorded as being carbapenem resistant in most countries [21]. Carbapenem resistance is omitted and underestimated in veterinary screening studies. This may be the result of its omission from the World Organisation for Animal Health (WOAH) List of Antimicrobial Agents of Veterinary Importance [22]. The European Medicine Agency included carbapenems in category A (“Avoid”); so, their use in veterinary medicine in the European Union (EU) is not authorized. There are exceptions for emergency clinical cases in companion animals, under the cascade according to Article 112 of the Veterinary Medicinal Products Regulation 2019 of the European Union Legislation [18].

Because of their close contact, the transmission of resistant strains of *P. aeruginosa* can occur between these species of animals and humans. Even in cases when chemotherapeutics are not in use in some species through the interspecies transmission of bacteria, the resistance determinants can also be transferred. In the current study, resistance to carbapenems was included in screening following the One Health Concept [19]. Moreover, carbapenem-resistant or multidrug-resistant *P. aeruginosa* strains are also equipped with other virulence factors, which directly influence the severity of infection [23]. Some of these are characteristic of acute or chronic infection. Pili, exotoxin S, exotoxin A, and phospholipase C are important for the acute phase of the disease, while siderophores (pyoverdin and pyochelin) and pseudo-capsules of alginate are essential for the chronic phase of *P. aeruginosa* infections [24]. Additionally, the characteristics of the secretion system of *P. aeruginosa*, being a type III secretion system, play an important role in the virulence of this bacteria. This system delivers virulence factors such as exotoxins S (*exoS*), T (*exoT*), U (*exoU*), and Y (*exoY*) [25]. *Pseudomonas* exotoxins have properties that enable invasion, colonization, and spreading in the host organism: *exoS* changes the function of the host cytoskeleton; *exoU* causes cell lysis and necroptosis in epithelial cells, macrophages, and neutrophils; *exoT* impairs the production of reactive oxygen species burst in neutrophils and promotes the apoptosis of host cells; *exoY* disrupts the actin cytoskeleton and increases endothelial permeability; and *toxA* inhibits protein synthesis [26]. Moreover, *P. aeruginosa* possesses other exoenzymes: *lasB* (Elastase B) has zinc metalloprotease activity and causes elastin degradation; *plcN* (non-haemolytic phospholipase C) releases phosphate esters from phosphatidylserine and phosphatidylcholine; and *plcH* (haemolytic phospholipase C) releases phosphate esters from sphingomyelin and phosphatidylcholine [26].

The current study aims to screen and compare the distribution of selected virulence factor genes and antibiotic resistance in *P. aeruginosa* strains of pet origin. The results of this study will help guide empirical antimicrobial selection for the treatment of dogs and cats infected with *P. aeruginosa* in veterinary medicine. In addition, the data generated from this study will contribute to antimicrobial resistance surveillance programmes in animal health.

## 2. Materials and Methods

### 2.1. Isolates

This study includes 271 *P. aeruginosa* isolates collected from cats (*n* = 59) and dogs (*n* = 212) from the Lower Silesia area between 2017 and 2020 in the commercial veterinary laboratory, VETLAB Ltd., in Wrocław, Poland. The bacterial isolates were transferred to the Department of Epizootiology and Clinic of Birds and Exotic Animals, Wrocław University of Environmental and Life Sciences. The isolates were cultured on the Columbia Blood Agar Base (OXOID, Basingstoke, UK) with 5% sheep blood and MacConkey Agar (OXOID, Basingstoke, UK). Then isolates were checked for oxidase production (Oxidase Detection Strips MICROGEN MID-61g GRASO BIOTECH, Starogard Gdański, Poland). Pure strains of nonfermenting bacilli, suspected to be *P. areuginosa* strains were frozen in 1 mL of Brain Heart Infusion Broth (BHI) (OXOID, Basingstoke, UK) supplemented with 30% glycerol and storage in −80 °C for up to six months for further analysis. Subsequently they were used for molecular investigation. Belonging to the *P. aeruginosa* species was provided by polymerase chain reaction (PCR). The details of procedure used for species identification of the *P. aeruginosa* strains used in the current study is described in Płókarz et al. (2022) [27]. The bacterial strains originated from pets with clinical manifestations of respiratory tract infections (swabs from the nasal cavity, conjunctival sacs, oral cavity, throat, trachea, and bronchi), skin, wounds, joint fluid, and genitourinary tract. Details are included in Figure 1 and Figure 2.

### 2.2. Antibiotic Resistance

All *P. aeruginosa* isolates were screened for antibiotic resistance using the Kirby–Bauer disc diffusion method. The concentration of bacterial suspensions was adjusted according to the 0.5 McFarland turbidity standard. The antibiotics and their corresponding amounts (μg/disc) used in antimicrobial disc diffusion tests are as follows: enrofloxacin (5), ciprofloxacin (5), ceftazidime (30), amikacin (30), gentamicin (10), tobramycin (10), ticarcillin (75), imipenem (10), meropenem (10) (OXOID, Basingstoke, UK), marbofloxacin (5), and pradofloxacin (5) (MAST Group, Liverpool, UK). Antimicrobial-resistant phenotyping of the isolates was performed and interpreted according to the Clinical and Laboratory Standards Institute document M100-S32 and VETO1S [28]. *P. aeruginosa* ATCC 27853 was used as a quality control strain.

### 2.3. Detection of Virulence Genes

The presence of toxin genes and occurrence of other virulence factors genes was assayed in DNA isolated from the *P. aeruginosa* strains. Isolation of DNA was carried out as previously described by Płókarz et al. (2022) [27]. The DNA was amplified in a thermocycler (Bio-Rad, Marnes-la-Coquette, France) in 2.5 µL buffer (MgCl_2_ at 20 mM concentration), 0.2 µL Taq DNA polymerase (5 U/L) (Thermo Fisher Scientific, Vilnius, Lithuania), 0.2 µL dNTP mix 10 mM concentration (Thermo Fisher Scientific, Vilnius, Lithuania), 0.2 µL of each specific primer, and 2 µL template DNA. The reaction volume was adjusted to 25 µL with sterile water. The used primers and cycle conditions are presented in Table 1.

### 2.4. Statistical Analysis

Statistical analysis was performed using TIBCO Statistica 13.3 (TIBCO Software Inc., Palo Alto, CA, USA). Spearman’s rank correlation coefficient was used to analyse the correlation between antibiotic resistance and the origin *P. aeruginosa* isolates Wilson’s test was used to establish the confidence interval (95% PU). The chi-square test and Fisher’s exact test were used to determine the significant difference in the virulence factor’s gene occurrence between *P. aeruginosa* strains susceptible and resistant to selected antibiotics. The statistical significance level was configured as *p* < 0.05.

## 3. Results

### 3.1. Antibiotic Resistance

In the current study, the resistance of 271 *P. aeruginosa* strains to specific antibiotics was investigated. In the beginning, the differences between the antibiotic resistance of *P. aeruginosa* originating from dogs or cats were tested. Only in the fluoroquinolone class significant differences between dogs and cats occur. In dogs, *P. aeruginosa* isolates were statistically significantly more often resistant to fluoroquinolone than isolates from cats (*p* ≤ 0.001). In other groups, there were no statistically significant differences. In *P. aeruginosa* strains from both dogs and cats, the most frequently detected resistance was to fluoroquinolones (from 17.3% (CI 95%: 13.3–22.3%) for ciprofloxacin to 83% (CI 95%: 78.1–87%) for enrofloxacin). Among the aminoglycosides, the highest resistance was observed to gentamicin at 26.9% (CI 95%: 22–32.5%), whereas in the carbapenem class, resistance to imipenem (14% (CI 95%: 8.2–15.8%)) was around two times higher than to meropenem (4.8% (CI 95%: 2.8–8%)). The detailed data are presented in Figure 3.

In the *P. aeruginosa* strains from dogs, there were some statistically significant correlations between the resistance to a particular antibiotic and the material from which bacteria was isolated; for example, the strains from wounds were frequently resistant to ticarcillin (*p* = 0.014), the strains from the urogenital system to meropenem (*p* = 0.042), the strains from the external auricular canals to marbofloxacin (*p* = 0.016), and from the skin and appendages to ciprofloxacin (*p* = 0.023) and gentamicin (*p* = 0.022). The detailed data are presented in Table 2. In cats, this correlation occurred only in the case of tobramycin, where strains from the respiratory system were significantly less resistant to this antibiotic (*p* = 0.026) (Table 3).

### 3.2. Virulence Factor Genes

Almost all *P. aeruginosa* strains harboured the *exoT* (97.8%) and *lasB* (93.4%) genes, while the lowest prevalence was found for the *exoU* (17.3%) and *plcH* (17.3%) genes. There was no observation of differences in the presence of exoenzyme genes in strains of dog or cat origin except for the occurrence of *exoS*, which was significantly more frequently detected in the *P. aeruginosa* from cats (*p* = 0.029). The detailed data are presented in Figure 4. There were only a few statistically significant associations between harbouring the virulence gene and exhibiting resistance to a particular antibiotic. In the *P. aeruginosa* strains from dogs, there were connections such as *toxA* and ceftazidime (*p* = 0.012) and *exoU* and marbofloxacin (*p* = 0.025) and amikacin (*p* = 0.056), while in the strains from cats, *exoU* and enrofloxacin (*p* = 0.054) were associated. The detailed data are presented in the Appendix A Appendix A.

## 4. Discussion

The disc diffusion method is still a widespread way to diagnose antibiotic resistance in commercial laboratories in Poland, because it is a relatively cheap and reliable method. Previous reports have indicated good effectiveness in the confirmation of resistance to antibiotics compared with determination of the minimal inhibitory concentration method (MIC) [31]. The disc diffusion method can be useful for screening antibiotic resistance. As our research considered a large number of *P. aeruginosa* strains under investigation, we decided to assess antibiotic resistance using this method.

In the current study, *P. aeruginosa* strains were most frequently resistant to fluoroquinolones. The majority of strains, 225 from among 271, presented resistance to enrofloxacin (83%), while the *P. aeruginosa* strains were the most susceptible to ciprofloxacin (only 47 from among 271 (17.3%) of *P. aeruginosa* were resistant). In bacteria originating from cats, the resistance to ciprofloxacin was low, at 3 from among 59 *P. aeruginosa* strains (5.1%), compared with the *P. aeruginosa* strains from dogs, where 44 from among 212 were resistant (20.8%). The obtained results are similar to those previously reported by Penna et al. (2011) [32], where 13.8% of strains of *P. aeruginosa* were found to be resistant to ciprofloxacin, while 63.6% were resistant to enrofloxacin. In the study by Eliasy et al. (2020), where the antimicrobial resistance patterns of *P. aeruginosa* from clinical samples from dogs were investigated, a high proportion of strains resistant to fluoroquinolones (enrofloxacin 73% and orbifloxacin 90%) were also detected [33]. Similar tendencies were also observed in strains of human origin where the resistance to ciprofloxacin was 31.36% and the resistance to levofloxacin was 79.57% [34]. However, we observed significantly lower resistance of *P. aeruginosa* strains to aminoglycosides compared with the previous study by Penna et al. (2011), where the resistance of strains was 71.4% for gentamicin and 64.4% for tobramycin [32]. Accordance with our results resistance to aminoglycosides was as follows: amikacin (16%), gentamicin (18%), and tobramycin (12%). The differences in the obtained results could be caused by the origin of the bacteria. In the earlier study, only *P. aeruginosa* strains from the causes of canine *otitis externa* were investigated [32], whereas, in the current study, we analysed the material from different pathological changes in pets. Considering also that aminoglycosides are frequently used in topical therapy for external ear canal infections, the increased resistance to these antibiotics is very probable. In contrast, very low levels of *P. aeruginosa* strains resistant to aminoglycosides were detected in Japan, where only 4.5% of strains exhibited resistance to gentamicin and 2.5% to amikacin [35].

There were also differences in resistance levels to ceftazidime, where 58% was reported by Penna et al. (2011) [32] and 77% by Eliasi et al. (2020) [33], while in the current study 21.7% of the *P. aeruginosa* strains from dogs and 16.9% from cats were detected as resistant. This is in contrast to other veterinary reports, where the prevalence of resistance to ceftazidime was 0–0.5% [35,36,37]. The resistance to ticarcillin (6.6% of *P. aeruginosa*) in the current study is very similar to the results obtained by Pottier et al. (2022) [36] in strains from veterinary samples (7.1%). In similar studies, the reported resistance of *P. aeruginosa* to carbapenems was consistent with our results 6–10% [33,38,39] to imipenem and 2–6% [36,38] to meropenem. While the results provided by Dégi et al. (2021) are significantly different to the results obtained in the current study [40]. In *P. aeruginosa* strains isolated from dogs in Western Romania, the resistance rate was 74.14% to meropenem and 77.59% to imipenem and was significantly higher than in the current study. The reason could be that samples in the mentioned research were collected the most frequently from superficial infections while our study shows mainly samples collected from external auricular canal [40]. The resistance to carbapenems in animal *P. aeruginosa* strains is still at the control level and is much lower than in the bacteria that cause nosocomial infections. Occurrence of resistance against this group of antibiotics in dogs and cats are probably results of close contact between owners and pets. It suggests exchanging genetic elements such as self-transmissible conjugative plasmids through horizontal gene transfer (HGT) [41]. The recent study of the antibiotic resistance of *P. aeruginosa* from Polish hospitals found that 67.8% of strains are resistant to imipenem and 29.2% to meropenem [42]. According to a 2017 ECDC survey, the average resistance rate to carbapenems (imipenem + meropenem) was 24.2% across Poland [43].

Our report also describes the prevalence of the chosen virulence factor genes in the *P. aeruginosa* strains under investigation. Overall, the frequency determined for the *exoU* (17.3%) and *exoS* (79,7%) genes is comparable to the results obtained by Hayashi et al. (2021) of 15.8% and 82.5%, respectively [38]. Similar to our results, *lasB* was detected in almost all *P. aeruginosa* (98.7%) [44]. The high prevalence of *exoT* and *exoY* genes in the current study (over 80% of *P. aeruginosa* strains) was consistent with previous reports, where all or almost all strains harboured such genes (100% *exoT* and 89–100% *exoY*) [45,46].

Our results show the presence of the *toxA* gene in 66.4% of strains, which is significantly lower than the 91.7% obtained by Hattab et al. (2021) [47]. Significant differences were also found for the prevalence of the *plcH* gene, with only 17.3% found to harbour the *plcH* gene in our study versus 91.7% in the earlier study. This difference could be caused by the very small number of samples (*n* = 24) compared with our study (*n* = 271), which included only canine samples. In our study, 59.8% of isolates were found to possess the *plcN* gene, which was lower than found by Aslataş et al. (2022) in clinical cases of bovine mastitis (100%), but they examined isolates from other anatomical locations and species, and the number of samples was significantly lower [48]. It is very difficult to compare the prevalence of *P. aeruginosa* isolates across different studies because of the differences in examined animal species, and geographical regions, all of which can exert varying influences on the results.

The association between the presence of *toxA* and the antibiotic resistance of *P. aeruginosa* strains found in our study is supported by the earlier results of Amormozafari et al. (2016), who used the same statistical analysis [49]. The presence of *exoU*, *exoS*, *exoT*, and *exoY* was analysed in humans by Horna et al. (2019). The results of their research confirm the association between the resistance to fluoroquinolone and aminoglycoside and the occurrence of the *exoU* gene [45]. In our research, we found a significant association indicating that the presence of the *exoU* gene in *P. aeruginosa* strains is linked with resistance to marbofloxacin (*p* = 0.025) and to amikacin (*p* = 0.056). For the other toxin genes, however, there were no statistically significant associations. The study performed by Hayashi et al. in 2021 in canine and feline patients from Japanese animal hospitals showed a significant association of isolates possessing *exoU* and resistance to aminoglycoside [38].

## 5. Conclusions

*Pseudomonas aeruginosa* isolates tested in the current study showed the highest resistance to fluoroquinolone. This can be useful in clinical veterinary practice in experimental antibiotic therapy and suggests that veterinarians should use greater caution in the selection of antibiotics. Of note is the low resistance to aminoglycosides in cases of *otitis externa*, despite commercial usage of this antibiotics class as a component of ear drops. Of considerable epidemiological value, our study included an antibiogram assessment of resistance to meropenem, imipenem, and ticarcillin in isolates from dogs and cats. In our knowledge carbapenems are not used in veterinary medical practice in Poland which suggests transferring *P.aeruginosa* resistant strains from human to animals. Furthermore, results of research showed a higher rate of resistance in humans which supports the thesis: humans are a source of *P.aeruginosa* resistance strains for animals. Although the rate of the resistant strains was still low, the increasing tendency indicates control of this potential reservoir is important. The analysis of the associations between the strains possessing virulence-factor-encoding genes and antibiotic resistance allows us to broaden the horizons considering antibiotic resistance. Further research on the relationships between virulence factors and antibiotic resistance in animals is required.

## Figures and Tables

**Figure 1 antibiotics-12-01226-f001:**
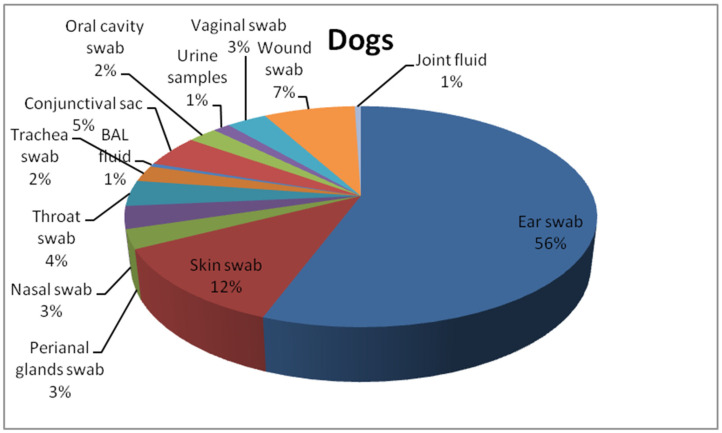
Collection sites of swabs or another sort of material where *P. aeruginosa* was isolated in dogs.

**Figure 2 antibiotics-12-01226-f002:**
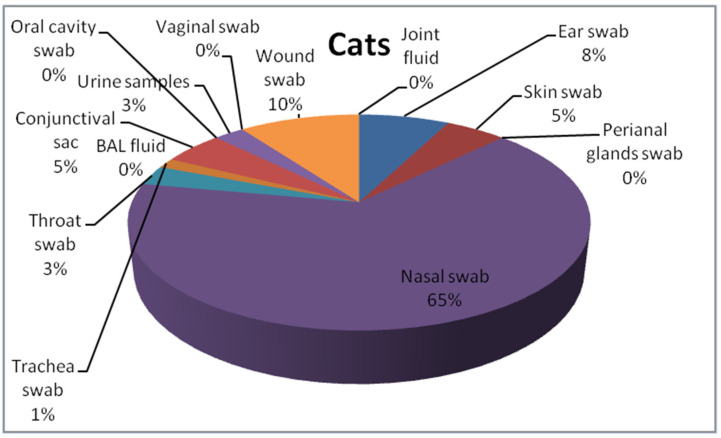
Collection sites of swabs or another sort of material where *P. aeruginosa* was isolated in cats.

**Figure 3 antibiotics-12-01226-f003:**
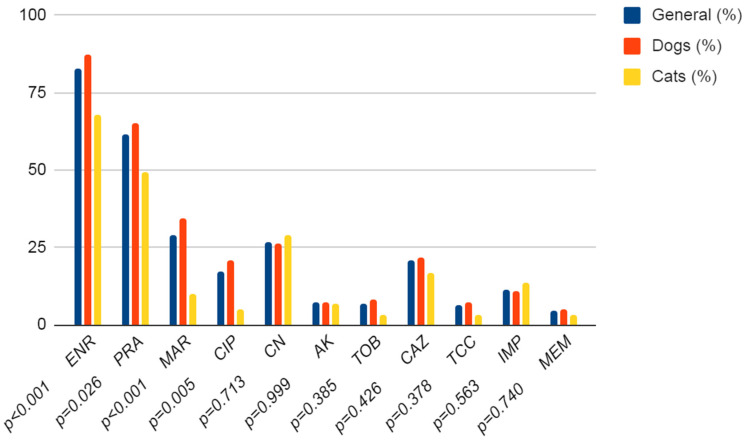
Antibiotic resistance of the *P. aeruginosa* strains originating from dogs and cats. ENR—enrofloxacin; PRA—pradofloxacin; MAR—marbofloxacin; CIP—ciprofloxacin; CN—gentamicin; AK—amikacin; TOB—tobramycin; CAZ—ceftazidime; TCC—ticarcillin; IMP—imipenem; MEM—meropenem, *p*—value.

**Figure 4 antibiotics-12-01226-f004:**
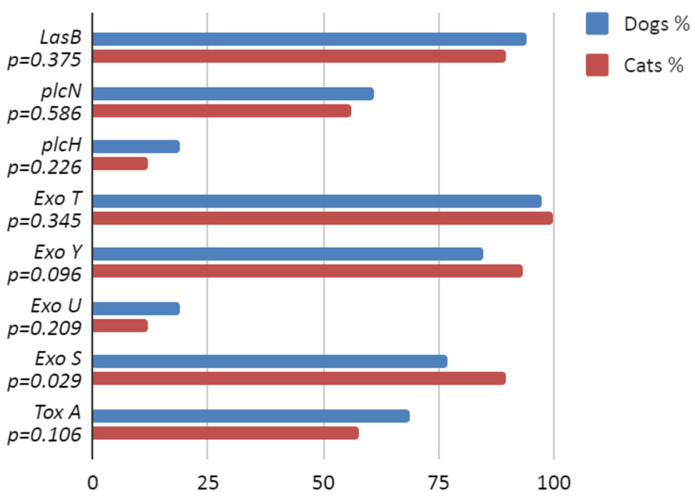
Virulence factor genes’ prevalence: comparison of *P. aeruginosa* isolates from dogs and cats. Genes encoding: LasB—elastase B, plcN—non-haemolytic phospholipase C, plcH—haemolytic phospholipase, exoT—exotoxin T, exoY—exotoxin Y, Exo U—exotoxin U, exoS—exotoxin T, tox A—exotoxin A.

**Table 1 antibiotics-12-01226-t001:** The primers and PCR assay conditions used in the current study.

Gen	Primers	Annealing Temperature	Product Size	References
Las B	F:5′-GGAATGAACGAAGCGTTCTCCGAC-3′ R:5′-TTGGCGTCGACGAACACCTCG-3′	55 °C ^a^	284	Wolska et al., 2009 [29]
plcN	F:5′-TCCGTTATCGCAACCAGCCCTACG-3′ R:5′-TCGCTGTCGAGCAGGTCGAAC-3′	55 °C ^a^	481	Wolska et al., 2009 [29]
plcH	F:5′-GCACGTGGTCATCCTGATGC-3′ R:5′-TCCGTAGGCGTCGACGTAC-3′	55 °C ^a^	608	Wolska et al., 2009 [29]
ToxA	F:5′-CTGCGCGGGTCTATGTGCC-3′ R:5′-GATGCTGGACGGGTCGAG-3′	55 °C ^a^	270	Wolska et al., 2009 [29]
ExoS	F:5′-CGTCGTGTTCAAGCAGATGGTGCTG-3′ R:5′-CCGAACCGCTTCACCAGGC-3′	55 °C ^a^	444	Wolska et al., 2009 [29]
ExoU	F:5′-CCGTTGTGGTGCCGTTGAAG-3′ R:5′-CCAGATGTTCACCGACTCGC-3′	58 °C ^b^	134	Mokhtaria and Amini 2019 [29]
ExoY	F:5′-CGGATTCTATGGCAGGGAGG-3′ R:5′-GCCCTTGATGCACTCGACCA-3′	58 °C ^b^	289	Mokhtaria and Amini 2019 [30]
ExoT	F:5′-AATCGCCGTCCAACTGCATGCG-3′ R: 5′-TGTTCGCCGAGGTACTGCTC-3′	58 °C ^b^	152	Mokhtaria and Amini 2019 [30]

^a^ 30 cycles of 30 s at 94 °C, 1 min at annealing temperature, and 1.5 min at 72 °C. ^b^ 36 cycles of 30 s at 94 °C, 30 s at annealing temperature, and 1 min at 72 °C.

**Table 2 antibiotics-12-01226-t002:** Correlation between the antibiotic resistance and the location of origin of the *P. aeruginosa* strains in dogs.

Antibiotic	External Auricular Canal (*n* = 118)		Skin and Appendages (*n* = 31)		Respiratory System and Oral Cavity (*n* = 36)		Uro-Genital System (*n* = 10)		Wounds (*n* = 16)		*p*
	*n*	% (95% PU)	*n*	% (95% PU)	*n*	% (95% PU)	*n*	% (95% PU)	*n*	% (95% PU)	
ENR	103	87.3 (80.1–92.1)	24	77.4 (60.2–88.6)	31	86.1 (713–93.9)	10	100 (72.2–100)	16	100 (8.6–100)	0.052
MAR	50	42.4 (33.8–51.4)	6	19.4 (9.2–36.3)	7	19.4 (9.8–35.0)	2	20.0 (5.7–51.0)	7	43.8 (23.1–66.8)	0.016 *
CIP	31	26.3 (19.2–34.9)	1	3.2 (0.6–16.2)	7	19.4 (9.8–35.0)	1	10.0 (1.8–40.4)	4	25.0 (10.2–49.5)	0.023 *
PRA	80	67.8 (58.9–75.6)	16	51.6 (34.8–68.0)	25	69.4 (53.1–82.0)	4	40.0 (16.8–68.7)	12	75.0 (50.5–89.8)	0.172
CAZ	32	27.1 (19.9–35.8)	2	6.5 (1.8–20.7)	6	16.7 (7.9–31.9)	2	20.0 (5.7–51.0)	4	25.0 (10.2–49.5)	0.089
AK	11	9.3 (5.3–15.9)	1	3.2 (0.6–16.2)	2	5.6 (1.5–18.1)	1	10.0 (1.8–40.4)	1	6.3 (1.1–28.3)	0.757
CN	40	33.9 (26.0–42.8)	3	9.7 (3.3–24.9)	7	19.4 (9.8–350)	1	10.0 (1.8–40.4)	5	31.3 (14.2–55.6)	0.022 *
TOB	11	9.3 (5.3–15.9)	2	6.5 (1.8–20.7)	1	2.8 (0.5–14.2)	0	0 (0–27.8)	3	18.8 (6.6–43.0)	0.225
TTC	8	6.8 (3.5–12.8)	0	0 (0–11.0)	1	2.8 (0.5–14.2)	2	20.0 (5.7–51.0)	4	25.0 (10.2–49.5)	0.014 *
IMP	11	9.3 (5.3–15.9)	2	6.5 (1.8–20.7)	4	11.1 (4.4–25.3)	4	40.0 (16.8–68.7)	2	12.5 (3.5–36.0)	0.150
MEM	7	5.9 (2.9–11.7)	0	0 (0–11.0)	0	0 (0–9.6)	2	20.0 (5.7–51.0)	1	6.3 (1.1–28.3)	0.042 *

* statistically significant differences. ENR—enrofloxacin; PRA—pradofloxacin; MAR—marbofloxacin; CIP—ciprofloxacin; CN—gentamicin; AK—amikacin; TOB—tobramycin; CAZ—ceftazidime; TCC—ticarcillin; IMP—imipenem; MEM—meropenem.

**Table 3 antibiotics-12-01226-t003:** Correlation between antibiotic resistance and origin place of *P. aeruginosa* strains in cats.

Antibiotic	Respiratory System (*n* = 49)		Others (*n* = 10)		
	*n*	% (95% PU)	*n*	% (95% PU)	*p*
ENR	32	65.3 (51.3–77.1)	8	80.0 (49.0–94.3)	0.476
MAR	5	10.2 (4.4–21.8)	1	10.0 (1.8–40.4)	0.999
CIP	2	4.1 (1.1–13.7)	1	10.0 (1.8–40.4)	0.433
PRA	23	46.9 (33.7–60.6)	6	60.0 (31.3–83.2)	0.452
CAZ	8	16.3 (8.5–29.0)	2	20.0 (5.7–51.0)	0.673
AK	3	6.1 (2.1–16.5)	1	10.0 (1.8–40.4)	0.535
CN	13	26.5 (16.2–40.3)	4	40.0 (16.8–68.7)	0.453
TOB	0	0 (0–7.3)	2	20.0 (5.7–51.0)	0.026 *
TTC	2	4.1 (1.1–13.7)	0	0 (0–27.8)	0.999
IMP	6	12.2 (5.7–24.2)	2	20.0 (5.7–51.0)	0.613
MEM	1	2.0 (0.4–10.7)	1	10.0 (1.8–40.4)	0.313

* statistically significant differences. ENR—enrofloxacin; PRA—pradofloxacin; MAR—marbofloxacin; CIP—ciprofloxacin; CN—gentamicin; AK—amikacin; TOB—tobramycin; CAZ—ceftazidime; TCC—ticarcillin; IMP—imipenem; MEM—meropenem.

## Data Availability

The datasets used and/or analyzed during the current study are available via e-mail from the corresponding author on reasonable request.

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
