# Peer review of "Screening for Antimicrobial Resistance and Genes of Exotoxins in Pseudomonas aeruginosa Isolates from Infected Dogs and Cats in Poland"

_antibiotics, 2023, doi:10.3390/antibiotics12071226_

Round 1

Reviewer 1 Report (Previous Reviewer 4)

This is my second time to review this manuscript. The author has answered all my comments, and the manuscript has been improved a lot.

Table 1: Lack of the information for 5'~3'.

Figure 1 and Figure 2: Give the discription for abbreviation.

Author Response

Authors’ response: 

Thank you for all the critical comments. The changes were marked in yellow. The missing descriptions in Table 1 and Figures (currently 3 and 4) were added.

Reviewer 2 Report (Previous Reviewer 2)

The reviewer's report for the manuscript antibiotics-2427457

It is an important study for veterinary medicine but also public health because it updates the data on antibiotic resistance of P. aeruginosa strains isolated from dogs and cats from various diseases, thus resulting in a short x-ray of the current situation in Poland, in this domain, with importance and impact on public health. The topic is one to consider for publication in the journal Antibiotics. However, many elements and aspects require significant improvements before being considered for publication. For the reviewer, it is difficult to follow in the absence of line numbering in the Word document, the presence of paragraphs, words crossed out in yellow, etc. I consider, from my point of view, that these are omissions unsuitable for this prestigious journal. Moreover, it denotes the lack of attention on the authors' part in the manuscript's preparation and drafting. In conclusion, my decision is a major revision with the correction and improvement of the expectations below:

In the title, the name of the bacterial species is written in italics.

Abstract

what is the motivation for using the Kirby Baurer method to detect antibiotic resistance? Has the reliability of this method allowed reliable results to be obtained?

toxA gene – it is written in italics

Introduction

P. aeruginosa is an opportunistic bacterium and very rarely causes infection in healthy individuals; however, in certain circumstances, it can directly threaten the health or the life of an individual [1]. - I recommend more current bibliographic sources.

For example:

Dégi, J., MoÈ›co, O. A., Dégi, D. M., Suici, T., MareÈ™, M., Imre, K., & Cristina, R. T. (2021). Antibiotic Susceptibility Profile of Pseudomonas aeruginosa Canine Isolates from a Multicentric Study in Romania. Antibiotics (Basel, Switzerland)10(7), 846. https://doi.org/10.3390/antibiotics10070846

Wood, S.J.; Kuzel, T.M.; Shafikhani, S.H. Pseudomonas aeruginosa: Infections, Animal Modeling, and Therapeutics. Cells 202312, 199. https://doi.org/10.3390/cells12010199

González-Alsina, A.; Mateu-Borrás, M.; Doménech-Sánchez, A.; Albertí, S. Pseudomonas aeruginosa and the Complement System: A Review of the Evasion Strategies. Microorganisms 202311, 664. https://doi.org/10.3390/microorganisms11030664

In nature, P. aeruginosa can exist in biofilm, attached to a surface or substrate, or in planktonic form, in which it can actively move via its flagellum. - requires references

The risk factors to humans associated with P. aeruginosa infections are chronic obstructive pulmonary disease, diabetes, cystic fibrosis, immunosuppression (after organ or bone marrow transplantation), severe kidney and liver failure, and multiorgan failure [4]. - requires more bibliographic sources.

Pseudomonas are naturally resistant to penicillin G, aminopenicillin, cephalosporin of the I and II generation, macrolides, tetracyclines, chloramphenicol, quinolones, sulphonamides, and trimethoprim [1], which narrows down the available clinical options, especially for veterinary patients. - requires more bibliographic sources.

The possibility of antimicrobial-resistant strain transfer from dogs and cats to humans should be an important point in international infectious disease control. -  requires more bibliographic sources.

In the current study, resistance to carbapenems was included in screening following the One Health Concept. - requires more bibliographic sources.

In the introduction, It would also be helpful to discuss the frequent isolation sites of P. aeruginosa in the case of dogs and cats and the connection with the importance of the author's current study. There is much information in the introductory part, but there is no connection between them; no debate regarding the connection between antibiotic resistance and the interspecific transfer of the resistance phenomenon. In dogs and cats, there are categories of tested antibiotics that are forbidden to be used in veterinary medicine. The transfer phenomenon should also be explained from animal to man and man to animal.

 Materials and methods

P. aeruginosa ATCC 27,853 was used as a quality control strain. - it will be corrected with ATCC 27853, without a comma.

The isolates were cultured on the Columbia Blood Agar Base (OXOID, Basingstoke, UK) with 5% sheep blood and MacConkey Agar (OXOID, Basingstoke, UK). Then isolates were checked for oxidase production (Oxidase Detection Strips MICROGEN MID-61g GRASO BIOTECH, Starogard GdaÅ„ski, Poland). Pure strains of P. areuginosa strains were frozen in 1 ml of Brain Heart Infusion Broth (BHI).... – is this isolation and identification methodology your own or has it been used in other studies? Only using Columbia blood agar and MacConkey, respectively the oxidase test, can we obtain pure colonies of P. aeruginosa?

Was the determination of Pseudomonas species based on the PCR method? Because in the study conducted by PÅ‚ókarz et al. (2022), mentioned in the manuscript, writes that: After pre-selection, isolates with P. aeruginosa identification were confirmed by polymerase chain reaction (PCR) based on amplification of two outer membrane lipoproteins genes, oprI and oprL.

Otherwise, the statement cannot be used: .... Pure strains of P. areuginosa strains.......

What were the arguments for choosing the antibiotics tested in the present study?

To avoid misunderstandings, I recommend rewording this chapter.

Results and discussion

The values ​​of p are written in italics throughout the manuscript.

The disc diffusion method can be useful for screening antibiotic resistance, especially in community-associated P. aeruginosa strains. As our research considered a large number of P. aeruginosa strains under investigation, we decided to assess antibiotic resistance using this method. - Do you think that the MIC method with the help of the VITEK 2 Compact device is not relevant? And the classical diffusometric method requires a lot of time and a lot of labor to perform. Why do you think that the Kirby Bauer method is used more in the case of studies with pathogens associated with certain communities?

The majority of strains presented resistance to enrofloxacin (83%), while the P. aeruginosa strains were the most susceptible to ciprofloxacin (only 17.3% of P. aeruginosa were resistant). - Maybe it would be helpful if you highlighted how many resistant strains, of how many, and the percentage values.

The resistance to carbapenems in animal P. aeruginosa strains is still at the control level and is much lower than in the bacteria that cause nosocomial infections. - Explain or elaborate on this statement. If its use is prohibited for pets, where does even this low percentage of resistance come from? Are there practicing veterinarians who use this antibiotic or is it of human origin?

It is very difficult to compare the prevalence of P. aeruginosa isolates across different studies because of the differences in examined species, origins, and latitudes, all of which can exert varying influence on the results. - what did you mean? That there are differences between the examined species, the origin of the animals studied? What do you mean when you say latitude? It would have been simpler if you had written that antibiotic resistance is influenced by the geographical region, the animal species, etc. It is a cumbersome wording, difficult to understand.

In our research, we found a significant association indicating that the presence of the exoU gene in P. aeruginosa strains is linked with resistance to marbofloxacin and to amikacin. - have you calculated p-values ​​or other statistical parameters?

Conclusions

Of considerable epidemiological value, our study included an antibiogram assessment of resistance to meropenem, imipenem, and ticarcillin in isolates from dogs and cats. - It is true, but it is not highlighted enough. Are antibiotics used in veterinary medical practice in Poland? What is the percentage of resistance in the human population to these antibiotics, to be able to explain where this resistance comes from, from animals or from humans?

Author Response

Thank you for all the critical comments. The changes were marked in green.

In the title, the name of the bacterial species is written in italics.

Authors’ response: 

It was written in italics. 

Abstract

what is the motivation for using the Kirby Baurer method to detect antibiotic resistance? Has the reliability of this method allowed reliable results to be obtained?

toxA gene – it is written in italics

Authors’ response: 

Chosen method to detect antibiotic resistance was motivated in the discussion section. Italics were added. 

Introduction

  1. aeruginosa is an opportunistic bacterium and very rarely causes infection in healthy individuals; however, in certain circumstances, it can directly threaten the health or the life of an individual [1]. - I recommend more current bibliographic sources.

For example:

Dégi, J., MoÈ›co, O. A., Dégi, D. M., Suici, T., MareÈ™, M., Imre, K., & Cristina, R. T. (2021). Antibiotic Susceptibility Profile of Pseudomonas aeruginosa Canine Isolates from a Multicentric Study in Romania. Antibiotics (Basel, Switzerland), 10(7), 846. https://doi.org/10.3390/antibiotics10070846

Wood, S.J.; Kuzel, T.M.; Shafikhani, S.H. Pseudomonas aeruginosa: Infections, Animal Modeling, and Therapeutics. Cells 2023, 12, 199. https://doi.org/10.3390/cells12010199

González-Alsina, A.; Mateu-Borrás, M.; Doménech-Sánchez, A.; Albertí, S. Pseudomonas aeruginosa and the Complement System: A Review of the Evasion Strategies. Microorganisms 2023, 11, 664. https://doi.org/10.3390/microorganisms11030664

In nature, P. aeruginosa can exist in biofilm, attached to a surface or substrate, or in planktonic form, in which it can actively move via its flagellum. - requires references

The risk factors to humans associated with P. aeruginosa infections are chronic obstructive pulmonary disease, diabetes, cystic fibrosis, immunosuppression (after organ or bone marrow transplantation), severe kidney and liver failure, and multiorgan failure [4]. - requires more bibliographic sources.

Pseudomonas are naturally resistant to penicillin G, aminopenicillin, cephalosporin of the I and II generation, macrolides, tetracyclines, chloramphenicol, quinolones, sulphonamides, and trimethoprim [1], which narrows down the available clinical options, especially for veterinary patients. - requires more bibliographic sources.

The possibility of antimicrobial-resistant strain transfer from dogs and cats to humans should be an important point in international infectious disease control. -  requires more bibliographic sources.

In the current study, resistance to carbapenems was included in screening following the One Health Concept. - requires more bibliographic sources.

Authors' response: 

Thank you for all your suggestions. We have added lacking reference positions.

In the introduction, It would also be helpful to discuss the frequent isolation sites of P. aeruginosa in the case of dogs and cats and the connection with the importance of the author's current study. There is much information in the introductory part, but there is no connection between them; no debate regarding the connection between antibiotic resistance and the interspecific transfer of the resistance phenomenon. In dogs and cats, there are categories of tested antibiotics that are forbidden to be used in veterinary medicine. The transfer phenomenon should also be explained from animal to man and man to animal.

Authors' response: 

We agree with your suggestions. We tried to add all the lacking information. 

 Materials and methods

  1. aeruginosa ATCC 27,853 was used as a quality control strain. - it will be corrected with ATCC 27853, without a comma.

The isolates were cultured on the Columbia Blood Agar Base (OXOID, Basingstoke, UK) with 5% sheep blood and MacConkey Agar (OXOID, Basingstoke, UK). Then isolates were checked for oxidase production (Oxidase Detection Strips MICROGEN MID-61g GRASO BIOTECH, Starogard GdaÅ„ski, Poland). Pure strains of P. areuginosa strains were frozen in 1 ml of Brain Heart Infusion Broth (BHI).... – is this isolation and identification methodology your own or has it been used in other studies? Only using Columbia blood agar and MacConkey, respectively the oxidase test, can we obtain pure colonies of P. aeruginosa?

Was the determination of Pseudomonas species based on the PCR method? Because in the study conducted by PÅ‚ókarz et al. (2022), mentioned in the manuscript, writes that: After pre-selection, isolates with P. aeruginosaidentification were confirmed by polymerase chain reaction (PCR) based on amplification of two outer membrane lipoproteins genes, oprI and oprL.

Otherwise, the statement cannot be used: .... Pure strains of P. areuginosa strains.......

What were the arguments for choosing the antibiotics tested in the present study?

To avoid misunderstandings, I recommend rewording this chapter.

Authors' response: 

We have modified the descriptions. The justification for choosing the antibiotics and chemotherapeutics tested in the current study is in the discussion section. 

Results and discussion

The values ​​of p are written in italics throughout the manuscript.

The disc diffusion method can be useful for screening antibiotic resistance, especially in community-associated P. aeruginosa strains. As our research considered a large number of P. aeruginosa strains under investigation, we decided to assess antibiotic resistance using this method. - Do you think that the MIC method with the help of the VITEK 2 Compact device is not relevant? And the classical diffusometric method requires a lot of time and a lot of labor to perform. Why do you think that the Kirby Bauer method is used more in the case of studies with pathogens associated with certain communities?

Authors' response: 

We have modified the descriptions and corrected the p throughout the manuscript. 

The majority of strains presented resistance to enrofloxacin (83%), while the P. aeruginosa strains were the most susceptible to ciprofloxacin (only 17.3% of P. aeruginosa were resistant). - Maybe it would be helpful if you highlighted how many resistant strains, of how many, and the percentage values.

Authors' response: 

We agree with the suggestion, the lacking information was included.

The resistance to carbapenems in animal P. aeruginosa strains is still at the control level and is much lower than in the bacteria that cause nosocomial infections. - Explain or elaborate on this statement. If its use is prohibited for pets, where does even this low percentage of resistance come from? Are there practicing veterinarians who use this antibiotic or is it of human origin?

Authors' response: 

We pointed to humans as a source of carbapenem-resistant P. aeruginosa strains and explained this in the “Discussion” section. 

It is very difficult to compare the prevalence of P. aeruginosa isolates across different studies because of the differences in examined species, origins, and latitudes, all of which can exert varying influence on the results. - what did you mean? That there are differences between the examined species, the origin of the animals studied? What do you mean when you say latitude? It would have been simpler if you had written that antibiotic resistance is influenced by the geographical region, the animal species, etc. It is a cumbersome wording, difficult to understand.

Authors' response: 

Thank you for the suggestion. It was changed. 

In our research, we found a significant association indicating that the presence of the exoU gene in P. aeruginosa strains is linked with resistance to marbofloxacin and to amikacin. - have you calculated p-values ​​or other statistical parameters?

Authors' response: 

Yes, we have calculated p-values based on the Chi-square test. P-values were added in this part of the text.

Conclusions

Of considerable epidemiological value, our study included an antibiogram assessment of resistance to meropenem, imipenem, and ticarcillin in isolates from dogs and cats. - It is true, but it is not highlighted enough. Are antibiotics used in veterinary medical practice in Poland? What is the percentage of resistance in the human population to these antibiotics, to be able to explain where this resistance comes from, from animals or from humans?

Authors' response: 

We highlighted that humans can be the source of P.aeruginosa resistance strains and explained it in the text.

Reviewer 3 Report (Previous Reviewer 1)

The current study screens the resistance of Pseudomonas isolates to antibiotics providing a genotypic screening to the related genes. This study has no significance and it is not suitable for publication in the current form. The main concern: what is the aim of this study, just screen the resistance in cats and dogs. The study lacks the aim and significance and even a real correlation between phenotypic and genotypic resistance. Could the authors explain why they choose these genes to screen? I may advise the authors to complete this work correlating the phenotypic and genotypic resistance and explaining the causes.  

Language is accepted

Author Response

Thank you for dedicated time to review our manuscript submitted to Anitbiotics.

We still believe that our publication is important for scientist who track antibiotic susceptibility of microorganisms isolated from pets, also for EU structures defining recommendations for veterinarians treating animals.

The results of our research and risk factors in the population of animals in city can be transferred to useful information that can be a recommendation for veterinarians and research centers conducting research on bacterial infections of humans and animals.

Reviewer 4 Report (New Reviewer)

Dear authors, the manuscript entitled "Screening for antimicrobial resistance and genes of exotoxins in Pseudomonas aeruginosa isolates from infected dogs and cats in Poland" is a well-written paper, and it provides important information on Pseudomonas aeruginosa prevalence and antimicrobial resistance  among dogs and cats in Poland.  For its interesting results, well explored,  the manuscript will be appreciated by readers. 

However, before publication,  it needs some minor revisions: 

 Keywords: authors should add "aeruginosa" after Pseudomonas;

MM_ 2.1 Isolates : authors should specify also here the origin, that is to say the kind of samples collected, since it is reported then in Results. 

MM_2.2 Antibiotic resistance: remove the comma from ATCC 27853 

Manuscript needs a minor English revision. 

Author Response

Thank you for all the critical comments. The changes were marked in blue.

However, before publication,  it needs some minor revisions: 

 Keywords: authors should add "aeruginosa" after Pseudomonas;

Authors’ response: 

It was added. 

MM_ 2.1 Isolates : authors should specify also here the origin, that is to say the kind of samples collected, since it is reported then in Results. 

Authors’ response: 

Figures containing information about the origin of the specimen were added in the ” Isolates” section.

MM_2.2 Antibiotic resistance: remove the comma from ATCC 27853 

Authors’ response: 

It was corrected. 

Round 2

Reviewer 2 Report (Previous Reviewer 2)

Dear authors,

I appreciate the effort made to respond to my comments as a reviewer, point by point. Certainly, in this way, the quality of the manuscript gained a lot by going through this major revision process. However, before suggesting consideration for publication, I recommend the introduction in the text of a bibliographic source that he suggested and which I do not find in the revised manuscript: Dégi, J., MoÈ›co, O. A., Dégi, D. M., Suici, T., MareÈ™, M., Imre, K., & Cristina, R. T. (2021). Antibiotic Susceptibility Profile of Pseudomonas aeruginosa Canine Isolates from a Multicentric Study in Romania. Antibiotics (Basel, Switzerland), 10(7), 846. https://doi.org/10.3390/antibiotics10070846.

Author Response

Dear Reviewer

we correct all suggestion

This manuscript is a resubmission of an earlier submission. The following is a list of the peer review reports and author responses from that submission.

Round 1

Reviewer 1 Report

The authors performed a phenotypic and genotypic screening of the P. aureginosa resistance among animal isolates. The research is aimless and has no benefit to readers. 

- What is the real aim behind this study, screening resistance in animals and its impact on humans, the authors did not clarify this point. OR, they screen resistance for animal health, I wonder if meropenem, ticarcillin, and other used antibiotics are prescribed for cats and dogs.

- The observed increased resistance is obvious against quinolones, what is the reason, the authors did not explore this point? did these antibiotics are highly prescribed to animals or humans

- The authors linked the resistance to the origin of infection, which is Ok, but they must link the phenotypic resistance to the screened genes, evaluating the incidence of these resistance genes to the phenotypic resistance. 

- The most important and critical, the authors study microbial resistance so they must link phenotypic resistance to resistance encoding genes, here, they link resistance to virulence which is totally wrong, especially since they did not show the resistance impact on the pathogenesis. The virulence genes could be detected in both resistant and susceptible strains as well. They must correlate phenotypic resistance to genotypic typing showing the most predominant mechanism of resistance. By the way, genes are written in italic.

- Figures

Author Response

Reviewer1: 

Author’s response

Thank you for all the critical comments. We would like to emphasize it was a screening study and we have focused on epidemiological value. We appreciate your advice and will consider it in our further study.

The authors performed a phenotypic and genotypic screening of  the P. aeruginosa resistance among animal isolates. The research is aimless and has no benefit to readers.

 - What is the real aim behind this study, screening resistance in animals and its impact on humans, the authors did not clarify this point. OR, they screen resistance for animal health, I wonder if meropenem, ticarcillin, and other used antibiotics are prescribed for cats and dogs.

 - The observed increased resistance is obvious against quinolones, what is the reason the authors did not explore this point? did these antibiotics are highly prescribed to animals or humans 

- The authors linked the resistance to the origin of infection, which is Ok, but they must link the phenotypic resistance to the screened genes, evaluating the incidence of these resistance genes to the phenotypic resistance. 

- The most important and critical, the authors study microbial resistance so they must link phenotypic resistance to resistance encoding genes, here, they link resistance to virulence which is totally wrong, especially since they did not show the resistance impact on the pathogenesis. The virulence genes could be detected in both resistant and susceptible strains as well. They must correlate phenotypic resistance to genotypic typing showing the most predominant mechanism of resistance. By the way, genes are written in italic. - Figures 1 

Reviewer 2 Report

Dear Authors,

It is an important subject with an immediate impact on practicing veterinarians in the current global context regarding antibiotic resistance and the One Health context. However, before it is proposed for publication, I believe things need to be clarified and corrected, which is why my proposal is a major revision after revising the manuscript.

I will detail my requirements and recommendations in the following:

Abstract

The aim of the research is to screen these P. aeruginosa strains for antibiotic resistance and the presence of selected virulence factor genes. This statement does not correspond to the title proposed for this article, where the authors mention the association between these factors (antibiotic resistance and virulence factors). I recommend correcting and standardizing the entire manuscript.

toxA gene and exoU gene it is written in italics; please correct it

Introduction

It is unnecessary to write the full name of Pseudomonas aeruginosa, because it was mentioned for the first time in the abstract. After this paragraph, the abbreviated version, P. aureus, can be used.

Pseudomonas aeruginosa (P. aeruginosa) is an oxidase-positive Gram-negative bacillus with minimal nutrient requirements, a feature that, combined with the possibility of its growth across a broad range of temperature (4-44℃), is recognized to contribute to its ubiquity (......). References will be inserted.

In nature, P. aeruginosa can exist in biofilm, attached to a surface or substrate, or in planktonic form, in which it can actively move via its flagellum (....). References will be inserted.

Nevertheless, due to its high resistance level to antimicrobials and disinfectants, P. aeruginosa is a major cause of nosocomial infections (7.1% in European countries). Please elaborate to understand the statement better: is the fact that it presents a high level of resistance to antibiotics and disinfectants a significant cause of nosocomial infections? Does it only cause nosocomial infections? Does the percentage of 7.1% refer to nosocomial infections in European countries in dogs and cats?

The risk factors to humans associated with P. aeruginosa infections are chronic obstructive pulmonary disease, diabetes, cystic fibrosis, immunosuppression (after organ or bone marrow transplantation), severe kidney and liver failure, and multiorgan failure [4]. I recommend rewording with information related to dogs and cats or removing it from the introductory part. The study refers to aspects that are already known to dogs and cats, not to the public health risk.

The diversity of the antimicrobial mechanism of P. aeruginosa causes difficulties in eradicating the bacteria from healthcare environments (.....). References will be inserted.

The outer membrane of P. aeruginosa contains a variety of porins that affect movement in and out of the cell and efflux pumps that can actively pump antimicrobials out (.....). References will be inserted.

Very important from a public health protection point of view is acquired resistance, such as via plasmids, and the development of mutations within existing genes, which result in altered function. Attention must be paid to additional resistances that can develop through changes in expression, such as those related to biofilm formation or tolerance [6]. In the case of P. aeruginosa, there is always a risk to public health. According to the European Food Safety Authority, Panel on Animal Health and Welfare, P. aeruginosa is classified, with >90% certainty, as one of the most relevant antimicrobial-resistant bacteria in the EU [7]. It will be reformulated in a shortened version or eliminated from the introductory part, being generalities that do not bring new information to the researched topic.

Carbapenems, such as imipenem and meropenem, are effective antibiotics for the treatment of P. aeruginosa infections [8]. These antibiotics are commonly used to treat infections caused by Pseudomonas aeruginosa in dogs and cats. Does the statement refer to dogs and cats?

In recent years, the rate of carbapenem resistance in P. aeruginosa has increased worldwide and has become of great concern, since it significantly restricts the therapeutic options for patients. Increased carbapenem resistance among these organisms has been found worldwide, with around 10% to 50% of P. aeruginosa isolates recorded as being carbapenem resistant in most countries [9]. Does this information refer to dogs and cats or humans? Please mention it in the sentences above if it refers to dogs and cats.

Carbapenem resistance is omitted and underestimated in veterinary screening studies (.....). References will be inserted.

This may be result of its omission from the OIE List of Antimicrobial Agents of Veterinary Importance [10]. The European Medicine Agency included carbapenems in category A (“Avoid”); so, their use in veterinary medicine in the European Union (EU) is not authorized. There are exceptions for emergency clinical cases in companion animals, under the cascade according to Article 112 of the Veterinary Medicinal Products Regulation 2019 of the European Union Legislation [11]. It will be reworded or removed from the introductory part. Providing more relevant information about antibiotic resistance and resistance factors in dogs and cats would be necessary.

Additionally, the characteristics of the secretion system of P. aeruginosa, being a type III secretion system, play an important role in the virulence of this bacteria (....). References will be inserted.

This system delivers virulence factors such as exoenzymes S (exoS), T (exoT), U (exoU), and Y (exoY) [14]. it is written in italics; please correct it

Pseudomonas exoenzymes have properties that enable invasion, colonization, and spreading in the host organism: exoS changes the function of the host cytoskeleton; exoU causes cell lysis and necroptosis in epithelial cells, macrophages, and neutrophils; exoT impairs the production of reactive oxygen species burst in neutrophils and promotes the apoptosis of host cells; exoY disrupts the actin cytoskeleton and increases endothelial permeability; and exoA inhibits protein synthesis [15]. The genes' names it is written in italics; please correct it

Las B (Elastase B) has zinc metalloprotease activity and causes elastin degradation; plcN (non-haemolytic phospholipase C) releases Antibiotics 2021, 10, x FOR PEER REVIEW 3 of 11 phosphate esters from phosphatidylserine and phosphatidylcholine; and plcH (haemolytic phospholipase C) releases phosphate esters from sphingomyelin and phosphatidylcholine [15]. The genes' names it is written in italics; please correct it

The current study aims to compare the distribution of virulence factor genes and antibiotic resistance in P. aeruginosa strains of pet origin. The study's purpose is unclear; several variants are written in the manuscript. I recommend uniforming. Otherwise, it is not understood, association study, comparison, or screening?

Materials and methods

The bacterial isolates were transferred to the Department of Epizootiology and Clinic of Birds and Exotic Animals, Wrocław University of Environmental and Life Sciences, where they were used for further investigation. By what means were the bacterial strains preserved?

All P. aeruginosa isolates were screened for antibiotic resistance using the Kirby– Bauer disc diffusion method. Would it not have been more appropriate to use the microdilution method for determining antibiotic resistance?

The antibiotics and their corresponding amounts (μg/disc) used in antimicrobial disc diffusion tests are as follows: enrofloxacin (5), ciprofloxacin (5), ceftazidime (30), amikacin (30), gentamicin (10), tobramycin (10), ticarcillin (75), imipenem (10), meropenem (10) (OXOID, Basingstoke, UK), marbofloxacin (5), and pradofloxacin (5) (MAST Group, Liverpool, UK). What do the numbers in parentheses represent? Would it be helpful to specify each antibiotic's resistant clinical breakpoint. Please specify the reasons for selecting these antibiotics for testing because some are not used in dogs and cats, except in certain well-specified situations.

Results

In P. aeruginosa strains from both dogs and cats, the most frequently detected resistance was to fluoroquinolones (from 17.3% (CI 95%: 13.3–22.3%) for ciprofloxacin to 83% (CI 95%: 78.1–87%) for enrofloxacin), but this finding was statistically significant only in dogs. Should the value of p be specified?

Among the aminoglycosides, the highest resistance was observed to gentamicin at 26.9% (CI 95%: 22–32.5%), whereas in the carbapenem class, resistance to imipenem (1.4% (CI 95%: 8.2–15.8%)) was around two times higher than to meropenem (4.8% (CI 95%: 2.8–8%)). Should the value of p be specified?

Discussion

Previous reports have indicated good effectiveness in the confirmation of resistance to antibiotics compared with determination of the minimal inhibitory concentration method (MIC) [20]. Please elaborate, to better understand these arguments? Historically, in vitro susceptibility testing was routinely performed by disk diffusion (Kirby-Bauer) method. The size of the growth-free zone determined whether the bacterium was considered to be susceptible, resistant, or intermediate to a particular antibiotic. Although a useful guide for selecting an effective antibiotic, Kirby-Bauer testing could not tell the clinician the exact antibiotic concentration needed to achieve a therapeutic result.

The disc diffusion method can be useful for screening antibiotic resistance, especially in community-associated P. aeruginosa strains (....). References will be inserted.

As our research considered a large number of P. aeruginosa strains under investigation, we decided to assess antibiotic resistance using this method. From my point of view, it is not a solid argument for this, automatic systems can also be used, for example, VITEK 2, which determines MIC.

In the current study, P. aeruginosa strains were most frequently resistant to fluoroquinolones. Do you have information regarding a correlation between the excessive use of these categories of antibiotics in dogs and cats in Poland?

Similar tendencies were also observed in strains of human origin where the resistance to ciprofloxacin was 31.36% and the resistance to levofloxacin was 79.57% [23]. It is precious information, but it would be of great value in this study if the strains of P. aeruginosa were those from dogs and cats, that is if we knew their origin.

However, we observed significantly lower resistance of P. aeruginosa strains to aminoglycosides compared with the previous study by Penna et al. (2011), where the resistance of strains was 71.4% for gentamicin and 64.4% for tobramycin [21]. Exista vreo corelatie intre consumul de aminoglicozide si reziatenta redusa, specific Poloniei? Overall, the frequency determined for the exoU (17.3%) and exoS (79,7%) genes is comparable to the results obtained by Hayashi et al. (2021) of 15.8% and 82.5%, respectively [27]. Similar to our results, lasB was detected in almost all P. aeruginosa (98.7%) [31]. The high prevalence of exoT and exoY genes in the current study (over 80% of P. aeruginosa strains) was consistent with previous reports, where all or almost all strains harboured such genes (100% exoT and 89-100% exoY) [32,33]. Gene names are written in italics. Likewise, in the rest of the manuscript.

Conclusions

I recommend including information about the virulence factors, about which nothing is specified, and the respective importance of the association with the phenomenon of antibiotic resistance. It would be interesting, and we also mention how it influences the decisions of the choice of antibiotics by practicing veterinarians.

Author Response

Thank you for all the critical comments. Below we attached the specific response to each comment. 

 Dear Authors, It is an important subject with an immediate impact on practicing veterinarians in the current global context regarding antibiotic resistance and the One Health context. However, before it is proposed for publication, I believe things need to be clarified and corrected, which is why my proposal is a major revision after revising the manuscript. I will detail my requirements and recommendations in the following:

Authors’ response:

Thank you for your positive comments. We have carefully studied all of the changes, and we agree with all of them.

 Abstract

 The aim of the research is to screen these P. aeruginosa strains for antibiotic resistance and the presence of selected virulence factor genes. This statement does not correspond to the title proposed for this article, where the authors mention the association between these factors (antibiotic resistance and virulence factors). I recommend correcting and standardizing the entire manuscript.

Authors’ response:

I have tried to clarify the aim of our study and corrected the title.

 toxA gene and exoU gene – it is written in italics; 

Authors’ response:

We have corrected this element.

Please correct it Introduction it is unnecessary to write the full name of Pseudomonas aeruginosa, because it was mentioned for the first time in the abstract. 

Authors’ response:

We have corrected this element.

After this paragraph, the abbreviated version, P. aureus, can be used. Pseudomonas aeruginosa (P. aeruginosa) is an oxidase-positive Gram-negative bacillus with minimal nutrient requirements, a feature that, combined with the possibility of its growth across a broad range of temperature (4-44℃), is recognized to contribute to its ubiquity (......). 

References will be inserted. In nature, P. aeruginosa can exist in biofilm, attached to a surface or substrate, or in planktonic form, in which it can actively move via its flagellum (....). 

References will be inserted. 

Authors’ response:

It was corrected in the text.

Nevertheless, due to its high resistance level to antimicrobials and disinfectants, P. aeruginosa is a major cause of nosocomial infections (7.1% in European countries). Please elaborate to understand the statement better: is the fact that it presents a high level of resistance to antibiotics and disinfectants a significant cause of nosocomial infections? Does it only cause nosocomial infections? Does the percentage of 7.1% refer to nosocomial infections in European countries in dogs and cats? 

Author’s response

This percentage of 7,1 % applies to humans.

The risk factors to humans associated with P. aeruginosa infections are chronic obstructive pulmonary disease, diabetes, cystic fibrosis, immunosuppression (after organ or bone 2 marrow transplantation), severe kidney and liver failure, and multiorgan failure [4]. 

I recommend rewording with information related to dogs and cats or removing it from the introductory part. The study refers to aspects that are already known to dogs and cats, not to the public health risk. 

Authors’ response:

We have removed the sentence.

The diversity of the antimicrobial mechanism of P. aeruginosa causes difficulties in eradicating the bacteria from healthcare environments (.....). 

References will be inserted. 

Authors’ response:

We have inserted references a bit further in the text to include all information from the source. So the next reference in the text concerns the part above.

The outer membrane of P. aeruginosa contains a variety of porins that affect movement in and out of the cell and efflux pumps that can actively pump antimicrobials out (.....). 

References will be inserted. 

Authors’ response:

We have inserted references a bit further in the text to include all information from the source. So the next reference in the text concerns the part above.

Very important from a public health protection point of view is acquired resistance, such as via plasmids, and the development of mutations within existing genes, which result in altered function. Attention must be paid to additional resistances that can develop through changes in expression, such as those related to biofilm formation or tolerance [6]. In the case of P. aeruginosa, there is always a risk to public health. According to the European Food Safety Authority, Panel on Animal Health and Welfare, P. aeruginosa is classified, with >90% certainty, as one of the most relevant antimicrobial-resistant bacteria in the EU [7]. It will be reformulated in a shortened version or eliminated from the introductory part, being generalities that do not bring new information to the researched topic. Carbapenems, such as imipenem and meropenem, are effective antibiotics for the treatment of P. aeruginosa infections [8]. These antibiotics are commonly used to treat infections caused by Pseudomonas aeruginosa in dogs and cats. Does the statement refer to dogs and cats? In recent years, the rate of carbapenem resistance in P. aeruginosa has increased worldwide and has become of great concern, since it significantly restricts the therapeutic options for patients. Increased carbapenem resistance among these organisms has been found worldwide, with around 10% to 50% of P. aeruginosa isolates recorded as being carbapenem resistant in most countries [9]. 

Does this information refer to dogs and cats or humans? Please mention it in the sentences above if it refers to dogs and cats. Carbapenem resistance is omitted and underestimated in veterinary screening studies (.....). References will be inserted. 

Authors’ response:

We have inserted references a bit further in the text to include all information from the source. So the next reference in the text concerns the part above.

This may be result of its omission from the OIE List of Antimicrobial Agents of Veterinary Importance [10]. The European Medicine Agency included carbapenems in category A (“Avoid”); so, their use in veterinary medicine in the European Union (EU) is not authorized. There are exceptions for emergency clinical cases in companion animals, under the cascade according to Article 112 of the Veterinary Medicinal Products Regulation 2019 of the European Union Legislation [11]. 

It will be reworded or removed from the introductory part. 3 Providing more relevant information about antibiotic resistance and resistance factors in dogs and cats would be necessary. 

Additionally, the characteristics of the secretion system of P. aeruginosa, being a type III secretion system, play an important role in the virulence of this bacteria (....). 

References will be inserted. 

Authors’ response:

We have inserted references a bit further in the text to include all information from the source. So the next reference in the text concerns the part above.

This system delivers virulence factors such as exoenzymes S (exoS), T (exoT), U (exoU), and Y (exoY) [14]. it is written in italics; please correct it 

Authors’ response:

We have corrected this element.

Pseudomonas exoenzymes have properties that enable invasion, colonization, and spreading in the host organism: exoS changes the function of the host cytoskeleton; exoU causes cell lysis and necroptosis in epithelial cells, macrophages, and neutrophils; exoT impairs the production of reactive oxygen species burst in neutrophils and promotes the apoptosis of host cells; exoY disrupts the actin cytoskeleton and increases endothelial permeability; and exoA inhibits protein synthesis [15]. The genes' names it is written in italics; please correct it Las B (Elastase B) has zinc metalloprotease activity and causes elastin degradation; plcN (non-haemolytic phospholipase C) releases Antibiotics 2021, 10, x FOR PEER REVIEW 3 of  11 phosphate esters from phosphatidylserine and phosphatidylcholine; and plcH (haemolytic phospholipase C) releases phosphate esters from sphingomyelin and phosphatidylcholine [15]. 

The genes' names it is written in italics; please correct it 

Authors’ response:

We have corrected this element.

The current study aims to compare the distribution of virulence factor genes and antibiotic resistance in P. aeruginosa strains of pet origin. The study's purpose is unclear; several variants are written in the manuscript. I recommend uniforming. Otherwise, it is not understood, association study, comparison, or screening? 

Authors’ response:

We have included information about screening study and unified the aim.

Materials and methods The bacterial isolates were transferred to the Department of Epizootiology and Clinic of Birds and Exotic Animals, WrocÅ‚aw University of Environmental and Life Sciences, where they were used for further investigation. By what means were the bacterial strains preserved? 

Authors’ response:

We have described this part of our laboratory work.  As “preserved” we meant “banked”.

All P. aeruginosa isolates were screened for antibiotic resistance using the Kirby– Bauer disc diffusion method. Would it not have been more appropriate to use the microdilution method for determining antibiotic resistance? 

Authors’ response:

We used this method because of numbers of samples and price of material.  Based on satisfied results of comparison between microdilution and disc diffusion method available in literature.

The antibiotics and their corresponding amounts (μg/disc) used in antimicrobial disc diffusion tests are as follows: enrofloxacin (5), ciprofloxacin (5), ceftazidime (30), amikacin (30), gentamicin (10), tobramycin (10), ticarcillin (75), imipenem (10), meropenem (10) (OXOID, Basingstoke, UK), marbofloxacin (5), and pradofloxacin (5) (MAST Group, Liverpool, UK). 

Authors’ response:

The numbers in parentheses means  (μg/disc), how it was mentioned in the beginning of the sentence. We used the antibiotics typical for animals but also important for humans because the pets could be a potential source of infection for their owners. 

What do the numbers in parentheses represent? Would it be helpful to specify 4 each antibiotic's resistant clinical breakpoint. Please specify the reasons for selecting these antibiotics for testing because some are not used in dogs and cats, except in certain wellspecified situations.

Results In P. aeruginosa strains from both dogs and cats, the most frequently detected resistance was to fluoroquinolones (from 17.3% (CI 95%: 13.3–22.3%) for ciprofloxacin to 83% (CI 95%: 78.1–87%) for enrofloxacin), but this finding was statistically significant only in dogs. 

Authors’ response:

It was added in the previous sentence. In dogs, P. aeruginosa isolates were statistically significantly more often resistant to fluoroquinolone than isolates from cats (p=<0.001).

Should the value of p be specified? Among the aminoglycosides, the highest resistance was observed to gentamicin at 26.9% (CI 95%: 22–32.5%), whereas in the carbapenem class, resistance to imipenem (1.4% (CI 95%: 8.2–15.8%)) was around two times higher than to meropenem (4.8% (CI 95%: 2.8–8%)). Should the value of p be specified? 

Authors’ response:

The p value was added to Fig. 1. 

Discussion Previous reports have indicated good effectiveness in the confirmation of resistance to antibiotics compared with determination of the minimal inhibitory concentration method (MIC) [20]. 

Please elaborate, to better understand these arguments? Historically, in vitro susceptibility testing was routinely performed by disk diffusion (Kirby-Bauer) method. The size of the growth-free zone determined whether the bacterium was considered to be susceptible, resistant, or intermediate to a particular antibiotic. Although a useful guide for selecting an effective antibiotic, Kirby-Bauer testing could not tell the clinician the exact antibiotic concentration needed to achieve a therapeutic result. 

The disc diffusion method can be useful for screening antibiotic resistance, especially in community-associated P. aeruginosa strains (....). References will be inserted. 

As our research considered a large number of P. aeruginosa strains under investigation, we decided to assess antibiotic resistance using this method.

 From my point of view, it is not a solid argument for this, automatic systems can also be used, for example, VITEK 2, which determines MIC. In the current study, P. aeruginosa strains were most frequently resistant to fluoroquinolones. 

Author’s response:

We have focused on the epidemiological value of this study. We were committed to examining large amounts of samples. In our case we would have to buy commercial plates for microdilution tests because we didn’t have an automatic system.

Do you have information regarding a correlation between the excessive use of these categories of antibiotics in dogs and cats in Poland? 

Author’s response:

Unfortunately we didn’t find reports about this correlation but based on veterinary practice we suppose it occurs.

Similar tendencies were also observed in strains of human origin where the resistance to ciprofloxacin was 31.36% and the resistance to levofloxacin was 79.57% [23]. It is precious information, but it would be of great value in this study if the strains of P. aeruginosa were those from dogs and cats, that is if we knew their origin. 5 However, we observed significantly lower resistance of P. aeruginosa strains to aminoglycosides compared with the previous study by Penna et al. (2011), where the resistance of strains was 71.4% for gentamicin and 64.4% for tobramycin [21]. Exista vreo corelatie intre consumul de aminoglicozide si reziatenta redusa, specific Poloniei? Overall, the frequency determined for the exoU (17.3%) and exoS (79,7%) genes is comparable to the results obtained by Hayashi et al. (2021) of 15.8% and 82.5%, respectively [27]. Similar to our results, lasB was detected in almost all P. aeruginosa (98.7%) [31]. The high prevalence of exoT and exoY genes in the current study (over 80% of P. aeruginosa strains) was consistent with previous reports, where all or almost all strains harboured such genes (100% exoT and 89-100% exoY) [32,33]. 

Gene names are written in italics. 

Authors’ response:

We have corrected this element.

Likewise, in the rest of the manuscript. Conclusions I recommend including information about the virulence factors, about which nothing is specified, and the respective importance of the association with the phenomenon of antibiotic resistance. It would be interesting, and we also mention how it influences the decisions of the choice of antibiotics by practicing veterinarians. 

Author’s response

Thank you for all the critical comments. I have tried to improve the manuscript, especially the part of the choice of antibiotics. 

Reviewer 3 Report

The manuscript entitled “Association between Pseudomonas aeruginosa antibiotic resistance and the presence of virulence-factor-encoding genes in isolates from dogs and cats in Poland” is interesting, relevant, and well written and should be included in the special issue “Antimicrobial Resistance and Zoonoses”.

The study was well conducted, the results are adequately treated, and are presented and discussed in a clear, logical and coherent way.

Therefore, the manuscript should be considered for publication after some minor corrections/revisions are made.

Author Response

Reviewer 3:

 The manuscript entitled “Association between Pseudomonas aeruginosa antibiotic resistance and the presence of virulence-factor-encoding genes in isolates from dogs and cats in Poland” is interesting, relevant, and well written and should be included in the special issue “Antimicrobial Resistance and Zoonoses”. The study was well conducted, the results are adequately treated, and are presented and discussed in a clear, logical and coherent way. Therefore, the manuscript should be considered for publication after some minor corrections/revisions are made. 

Authors’ response:

Thank you for your positive comments. We have carefully studied all of the changes, and we agree with all of them. We have corrected all elements suggested by the reviewer and changed Keywords.

More specific comments: 

Abstract: to agree with the tense used in the rest of the abstract, the sentence " The aim of the research is to screen these ..." should be changed to " The aim of the research was to screen these ..."

 Keywords: the choice of key words does not seem right to me since it does not reflect the content of the manuscript and the word resistance is misspelled (change "reistance") 

Introduction (page 2; 2nd paragraph): “This may be result of its omission from the OIE List of ...” ought to be replaced by “This may be result of its omission from the World Organisation for Animal Health (OIE) List of ...” since the full name of this organization (WOAH, founded as Office International des Epizooties - OIE) had not yet been mentioned. 

Materials and methods: 

(page 3; last paragraph) - please put the number 2 in "MgCl2" in subscript (i.e. MgCl2) (page 4; table legend) - The indications given for annealing temperature for PCR in the notes of Table 1 (a and b, respectively) are the same, so it is not clear why the distinction is made 

(page 4; last paragraph) – resistance to imipenem is 14% and not “1.4%” was indicated; please change

 Results: (page 5) - Figure 1 is shown but there is no reference to it in the text; please correct.

 (page 5) – Table regarding correlation between the antibiotic resistance and the location of origin must be numbered as "2" (there is already a table 1 previously displayed). 

Please change in the text and in the title of the table. Additionally, in this table whenever correlation is significant, arrows appear but there is no explanation/reason associated with the fact.

 Finally, the table should be uncropped (on 2 pages).

 (page 6) For the same reason, the Table indicated as number 2 (regarding correlation between antibiotic resistance and origin place of P. aeruginosa strains in cats) must be renumbered to 3 (correct, also, in the text and in the title of the table).

 (page 6; last line) – “... in Fig.2.” should be altered to “... in Figure 2.” 

2 (page 7) – the sentence “The highest rate of resistances found in the strains under investigation was to enrofloxacin (83%), while the P. aeruginosa strains were the most susceptible to ciprofloxacin (only 17.3% of P. aeruginosa were resistant).” it is not well written (and as it stands it is confusing and contradictory); please rewrite 

(page 7) – the sentence “This is in accordance with our results for resistance to aminoglycosides (amikacin (16%), gentamicin (18%), and tobramycin (12%)). ” makes no sense at this position in the text (where fluoroquinolones were being addressed) so it must be misplaced. 

Discussion: (page 8) – “... versus. 91.7%in ...” should be altered to “... versus 91.7% in ...”. (page 8) – remove the word “also” in the sentence “... which also included only canine samples.”

 (page 8) – consider altering “The results of their research confirm the association between fluoroquinolone resistance and the occurrence of the exoU gene and between aminoglycoside resistance and the exoU gene.” to “The results of their research confirm the association between the resistance to fluoroquinolone and aminoglycoside and the occurrence of the exoU gene.” 

(page 8) –the title “Conclusion” should be placed on the next page. 

References: reference 22 is not properly formatted 1 

Author’s response:

Thank you for all specific suggestions. We have incorporated them in the text. 

Reviewer 4 Report

The results of this study have some significance that can help guide empirical antimicrobial selection for the treatment of dogs and cats infected with P. aeruginosa in veterinary medicine. However, there still have some more work need to do in the future, for example, in vitro antibiotics resistance analysis for the isolated P. aeruginosa not just based on correlation analysis; in vivo antibiotic resistance in animals.

Abstract: The number of decimal places of the p-value needs to be unified, for example, p= 0.054.

Figure 1 should be pointed out when describing the results.

In Figure 1 and Figure 2, there need to indicate significance.

In the legends of Figure 1 and Figure 2, the Abbreviations need to be explained.

When P. aeruginosa isolates collected from cats and dogs, if the animal species factors are taken into account? If animal species factor will influence the antibiotic resistance of P. aeruginosa?

The method for P. aeruginosa isolated need introduced in the Materials and methods.

Author Response

 The results of this study have some significance that can help guide empirical antimicrobial selection for the treatment of dogs and cats infected with P. aeruginosa in veterinary medicine. However, there still have some more work need to do in the future, for example, in vitro antibiotics resistance analysis for the isolated P. aeruginosa not just based on correlation analysis; in vivo antibiotic resistance in animals.

 Abstract: The number of decimal places of the p-value needs to be unified, for example, p= 0.054. 

Author’s response

We have  unified p-value.

Figure 1 should be pointed out when describing the results.

 In Figure 1 and Figure 2, there need to indicate significance.

Author’s response

We have added significance in Figure 1 and Figure 2.

 In the legends of Figure 1 and Figure 2, the Abbreviations need to be explained. 

Author’s response

We have inserted Abbreviations explaining.

When P. aeruginosa isolates collected from cats and dogs, if the animal species factors are taken into account? If animal species factor will influence the antibiotic resistance of P. aeruginosa?

Author’s response

We have included results in context animal species factors.

 The method for P. aeruginosa isolated need introduced in the Materials and methods.

Author’s response

We have described the method in the appropriate section.